# Testosterone and Prolactin Perturbations Possibly Associated with Reduced Levels of β-Arrestin1 in Mononuclear Leukocytes of Women with Premenstrual Dysphoric Disorder

**DOI:** 10.3390/ijms242015449

**Published:** 2023-10-22

**Authors:** Sanket Nayyar, Anthony Archibong, Tultul Nayyar

**Affiliations:** Meharry Medical College, 1005 Dr. D. B. Todd Jr. Blvd, Nashville, TN 37208, USA; snayyar@whiteriverhealth.org (S.N.); aarchibong@mmc.edu (A.A.)

**Keywords:** β-Arrestin, mood disorders, mononuclear leukocytes, PMDD, testosterone, prolactin, women

## Abstract

Previously, we reported that a reduction in β-Arrestin1 protein levels in peripheral blood mononuclear leukocytes (PBMC) significantly correlated with the severity of depression symptoms in women with premenstrual dysphoric disorder (PMDD). This study aimed to determine whether the reduced premenstrual β-Arrestin1 protein levels were associated with changes in the regulator for late luteal phase progesterone secretion. The study participants (*n* = 25) were non-pregnant women between 18 and 42 years of age not taking any antidepressants or receiving therapy and experiencing the luteal phase of menstruation. ELISA determined the β-Arrestin1 protein in PBMC; testosterone and prolactin levels from the plasma were determined by radioimmunoassay. Reduced levels of β-Arrestin1 protein in women with Hamilton Rating Scale for Depression (HAM-D) scores above 19 were observed alongside significantly higher plasma testosterone and prolactin concentrations. Understanding the mechanism underlying the initiation of PMDD will allow for identification of a key perturbed metabolic enzyme that can serve as a target for drug development to ensure the alleviation of PMDD, which has been suggested earlier as a risk factor for developing major depressive disorders.

## 1. Introduction

Menstruation, a physiological event that is experienced for 13–54 years in women of reproductive age in Western cultures [1,2] is accompanied by premenstrual disorders—namely premenstrual symptoms [PMS] and premenstrual dysphoric disorder [PMDD]—in 12% of the above-mentioned population [3].

In affected populations, PMS exhibits somatic and psychological symptoms during the progesterone-(P_4_)-dominated luteal phase of the menstrual cycle, which culminates in extreme distress and functional impairment [4]. Interestingly, PMS symptoms resolve within a few days of the onset of menstruation [5]. It has been documented that approximately 40–50% of women exhibit severe symptoms of PMS that interfere with daily routines [4]. Commonly reported symptoms of PMS are categorized as psychological, behavioral, and physical, such as those of PMDD and detailed in Kwan et al. [6].

Five percent of women diagnosed with premenstrual disorders have PMDD. This disorder is characterized by depression, feelings of hopelessness, anxiety and tension, mood swings, irritability, absent-mindedness, fatigue, increased appetite, sleep disturbance, feelings of overwhelm, poor coordination, headache, aches, bloating, and breast tenderness [6]. Treatment for premenopausal disorders includes the use of non-pharmaceutical and pharmaceutical agents for the alleviation of PMDD symptoms. Non-pharmaceutical therapies constitute the initial management strategy for women with mild symptoms of premenstrual disorders. These include changes in diet, exercise, cognitive behavioral treatment, acupuncture, etc. [7,8]. Pharmaceutical therapy for women with severe symptoms of premenstrual disorders (PMDD) include: (1) selective serotonin reuptake inhibitors (SSRIs) treatments; (2) combined treatment with different regimens of ethynyl estradiol and progestogen; (3) gonadotropin releasing hormone (GnRH) and estradiol and progestogen add-back (reviewed in Yonkers et al. [9]). 

The cause of premenstrual disorder is not well understood, but the onset of its symptoms is associated with ovarian hormone levels. Consequently, it has been proposed that symptoms of PMS occur simultaneously with the altered ratio of estrogen to P_4_ [10]. Symptoms are also linked to serotonin as a key etiological factor [11].

Women with PMS are hypersensitive to the normal pattern of ovarian steroids during the menstrual cycle [12]. Symptoms often get resolved during pregnancy, after menopause, or as a result of treatments with ovulation blockers (combined oral contraceptives) which suggest the involvement of cyclic ovarian steroids [9,13]. Unfortunately, the exact role of ovarian steroids in the resolution of symptoms of PMS is not universally consistent. Women on high doses of steroid contraceptives developed depression, while those on lower doses of steroid contraceptives did not [14]. On the contrary, the placement of women with no history of depression on hormonal contraceptives resulted in a greater risk of developing depression than their control counterparts [15], a phenomenon which prompts the cessation of hormonal contraceptive use by the affected subjects [16,17]. Further, some studies show no association between oral contraceptive use and depression symptoms, while others suggest that hormonal contraception is associated with better mood [18,19,20]. Notwithstanding the abovementioned inconsistencies, the above studies suggest some gonadal steroid involvement in the development of PMS/PMDD.

It is common practice to determine whether the stage of the menstrual-cycle-dependent estrogen and P_4_ ratios are altered when addressing symptoms of premenstrual disorders. Similarly, PMS may also be attributed to altered neuroendocrine functions; thus, ovarian steroid and neurotransmitter milieu should be considered together in evaluating women for premenstrual disorders. It is well documented that neurotransmitters (e.g., serotonin, dopamine norepinephrine, and gamma-amino butyric acid [GABA]) are important regulators of mood and well-being [21,22]. As estrogen increases the concentrations of serotonin [23], it is conceivable that fluctuating moods in women with premenstrual disorders may be linked to altered estrogen and serotonin secretion patterns. The increase in estrogen levels in women of reproductive age has been shown to be accompanied by a concomitant increase in serotonin during the follicular phase of the menstrual cycle [24]. Because serotonin improves mood [21,22], estrogen acts as an antidepressant and mood stabilizer under normal physiologic conditions. On the contrary, the luteal phase of the menstrual cycle is characterized by P_4_ dominance and a significant decline in estrogen secretion. Consequently, the drop in estrogen levels that occurs during the premenstrual stage of the cycle may serve as a trigger that lowers serotonin levels below functionality [25], ultimately resulting in mood swings as soon as estrogen levels rise again during the follicular phase of the new cycle.

In women on P_4_-only oral contraceptives (POCs), the highly expressed β-Arrestin protein in neural and immune systems [26] was suppressed [27] compared with their counterparts on P_4_ + estrogen oral contraceptives (COCs) or on no contraceptives (NCs) [27]. Further, the suppression of β-Arrestin in women on POC was followed concomitantly by a significantly higher Hamilton Rating Scale for Depression (HAM-D) than in women on COC or NC [27].

High physiological P_4_ doses have been shown to significantly inhibit monocyte release of proinflammatory agents, including β-Arrestin [27,28], suggesting that P_4_-induced immune suppression may contribute to depression [28]. Therefore, the peri-menstrual absence of estrogen and persistent P_4_ dominance can contribute to PMS/PMDD by significantly inhibiting P_4_-induced β-Arrestin1 (immune) suppression [27,28,29]. The present study is justified by data that show that estrogen induces P_4_ receptor synthesis and high P_4_ desensitizes its receptors [30,31]. These findings suggest that reduced conversion of androgens to estrogen and the contributing effect of increased androgen-induced elevated prolactin (PRL) on P_4_ increase during the peri-menstrual period may contribute to PMS/PMDD [32].

Our previous study [33] reported that measurement for β-Arrestin1 protein levels in peripheral blood mononuclear leukocytes (PBMC) could aid in distinguishing between PMS and PMDD. We observed that the Hamilton Rating Scale for Depression (HAM-D) scores and the magnitude of the different parameters on the HAM-D were significantly higher (Figure 1A,B), and β-Arrestin1 protein levels in PBMCs were significantly lower in women with PMDD compared with PMS women (Figure 2). The reduction in β-Arrestin1 protein levels was significantly correlated with the severity of depressive symptoms (Figure 3). Continuing our previous study [33], this study aimed to determine whether reduced β-Arrestin1 protein levels are associated with changes in the regulators for premenstrual P_4_ secretion.

## 2. Results

### 2.1. Analysis of Different Parameters on the Hamilton Rating Scale for Depression (HAM-D)

Twenty-five women of reproductive age (n = 12 of HAM-D > 19 depression group and n = 13 of HAM-D < 19 non-depression group) participated in this study. The severity of the depression symptoms in the two categories of women is indicated in the overall HAM-D scores in Figure 1A. All the parameters on the HAM-D scale (Figure 1B) were significantly greater in the depression group of women with an HAM-D score above 19 (PMDD) compared with that of the non-depression group whose HAM-D score was below 19.

### 2.2. Change in β-Arrestin1 Protein Concentrations Due to Depression

The concentrations of extracted β-Arrestin-1 protein from peripheral blood mononuclear leukocytes (PBMC) of the depression group were reduced (*p* < 0.004) by about 22.2% compared with those of their non-depression counterparts (Figure 2).

### 2.3. Correlation between β-Arrestin1 Protein Concentrations and HAM-D

By simple linear regression analysis, HAM-D scores were found to be negatively correlated with the concentrations of β-Arrestin1 proteins extracted from PBMC of participants (R = −0.63, *p* < 0.0009; Figure 3). This observation substantiates the prediction that individuals with high PBMC β-Arrestin1 protein concentrations are less prone to increases in HAM-D scores and, thus, to developing PMDD.

### 2.4. Depression-Associated Changes in Hormones That Regulate Steroidogenesis at the Perimenstrual Stage of the Menstrual Cycle Studied

Sera from women in the depression group contained higher (*p* < 0.001) concentrations (ng/mL) of testosterone (T) compared to sera from the women in the non-depression group (Figure 4). An increase of approximately 50% in the magnitude of depression-induced T concentrations was found among the patients in the depression group.

### 2.5. Correlation between Testosterone Concentrations and β-Arrestin1

Interestingly, there was no correlation between serum T and PBMC β-Arrestin1 concentrations in this study (Figure 5; R = 0.22; *p* > 0.05), suggesting that high serum concentrations of T were not predictive of the pattern of β-Arrestin1 secretion by PBMC.

Serum PRL concentrations (ng/mL) were approximately 103% higher (*p* < 0.0001) in the depression group (Figure 6) compared to those in the non-depression group.

Linear regression analysis showed that PRL concentrations did not correlate with the concentrations of β-Arrestin1 proteins in the PBMC of participants (Figure 7; R = 0.19; *p* > 0.05). This observation suggests that high circulating levels of PRL were not predictive of the pattern of β-Arrestin1 secreted by PBMC.

Further, regression analysis also revealed the absence of a correlation between serum PRL concentrations and the HAM-D scores of participants (upper segment: Pearson R value = −0.5; *p* = 0.08; Spearman R value = −0.26; *p* = 0.39; lower segment: Pearson R value = 0.19; *p* = 0.55; Spearman R value = 0.27; *p* = 0.42; confidence interval for all regression analyses was 95%; Figure 8). This observation suggests that high PRL secretion in female subjects did not directly influence HAM-D scores and vice versa.

## 3. Discussion

The higher values of the HAM-D scores during the luteal phase of the menstrual cycle recorded in patients from the depression group compared to patients in the non-depression group suggest an involvement of a constant pattern of P_4_ secretion in the PMS/PMDD subjects in this study. The involvement of continuous P_4_ in the induction of PMS/PMDD agrees with data generated by Smith et al. [27]. These data indicate that POC usage constitutes a potential risk factor for developing depression symptoms, while usage of COC or no contraceptives (NC) does not [27]. P_4_, secreted by the ovaries, have an immunosuppressive function [34]. The β-Arrestin protein secretion by monocytes is related to the immune status in the body [27,28]. According to Su [35], luteal P_4_ suppresses the innate immune system via inhibition of Toll-like receptor-4 (TLR-4) and Toll-like receptor-9 (TLR-9) and activation of interleukin-6 (IL-6) and nitric oxide (NO) production by macrophages. Hence, the reduction in β-Arrestin protein secretion and the negative correlation between β-Arrestin protein concentrations and HAM-D scores suggest a suppression of the immune system in the depression group, but not in the non-depression group [27,34,35]. This suggests that the secretion of P_4_ during the premenstrual period at levels comparable to those typical of the early- and mid-secretory phases of the cycle may contribute to the development of PMDD in the depression group. Data in this study also show a significant increase in the concentrations of T among subjects in the depression versus non-depression group. Similar data on luteal T profile in depressed women were reported by Lombardi et al. [36] and Ekholm et al. [37], indicating a possible involvement of androgens in the pathophysiology of premenstrual irritability and dysphoria. The lack of correlation between serum T concentrations and PBMC β-Arrestin-1 protein concentrations suggests inhibition of T conversion to estrogen during the perimenstrual period in women with PMDD.

In this study, the concentrations of PRL in the depression group during the premenstrual period were significantly higher compared to those recorded in patients from the non-depression group, and they were comparable to values reported by Elgellaie et al. [38]. Increased PRL among subjects in the depression group could indicate reduced inhibition of PRL secretion by central dopamine, suggesting an underlying pathophysiology of diminished dopamine neurotransmission as a cause of the depression symptoms [39]. The contribution of increased concentrations of PRL to the depression symptoms is further confirmed by the moderate negative correlation between β-Arrestin protein concentrations and HAM-D scores and by the absence of a correlation between PRL protein levels and HAM-D scores observed in this study. Conceivably, the negative correlation between β-Arrestin concentrations and HAM-D scores is a reflection of PRL-induced elevated P_4_, which ultimately inhibits the innate immune system via inhibition of TLR4- and TLR9-triggered β-Arrestin protein by macrophages [35]. The endocrine data in this study suggest that any deviation from the secretory pattern of T and PRL characteristic of those in the non-depression group may be related to depression symptoms observed in women in the susceptible group. However, it is not known whether the altered T and PRL secretion reflects a cause or effect of depression.

Interpretation of the significance of the altered changes in T and PRL in the depression group is surrounded by the lack of knowledge of the precise stage at which depression symptoms are triggered during the luteal phase of the menstrual cycle. Peripheral androgens (T, androstenedione, and dehydroepiandrosterone) in women are produced in response to stress by the adrenal cortex, with stress increasing PRL secretion [32]. It is conceivable that increased stress stimulated elevated luteal PRL among women in the depression group and, consequently, increased premenstrual P_4_ levels which then set the stage for the development of depression symptoms.

It has been shown that allopregnanolone (a neuroactive steroid) modulates neurotransmission [40]. Pregnanolone is produced in the brain, adrenals, and male and female gonads via a two-step metabolism of P_4_. Initially, P_4_ is metabolized into 5α-dihydroP_4_ (5α-DHP) by 5α-reductase which is then converted into 3α-5α-tetrahydroP_4_ (allopregnanolone [41]). According to Kimball et al. [40], less P_4_ is converted into allopregnanolone as the cycle progresses from the estrogen-dominated follicular phase to the P_4_-dominated luteal phase of the menstrual cycle. Arguably, among women with premenstrual disorders, the decrease in the allopregnanolone/P_4_ ratio may have been altered due to the saturation of 5α-reductase, thus leading to the maintenance of high P_4_/allopregnanolone ratio at the premenstrual phase of the cycle. This line of argument is supported by the study of Schüle et al. [42], which showed that reduced synthesis and secretion of allopregnanolone is associated with major depression and PMDD.

Further, it has been demonstrated that depression is associated with deficits in the dopaminergic system, the origin of which resides in the dysregulation of its regulatory afferent circuits [42]. If the deficits of the dopaminergic system are substantial, PRL concentrations from the pituitary gland of women in the depression group may increase significantly and contribute to increased T secretion as described above [36,37].

In conclusion, our study showed a significant decrease in PBMC β-Arrestin1 protein concentrations and a significant negative correlation between β-Arrestin1 proteins and HAM-D scores during the premenstrual stage of the cycle in patients belonging to the depression group. Further, we observed a substantial increase in serum T and PRL concentrations, as well as the absence of correlation between PRL concentrations and HAM-D scores during the premenstrual stage of the cycle in patients belonging to the depression group. More extensive studies are needed to investigate the pattern of ovarian and adrenal steroids across the menstrual cycle in healthy versus depressed women to determine the precise stage at which dysregulation leads to disorders in allopregnanolone synthesis. Also needed are further studies to explain the relationship between cortisol and allopregnanolone activity in the brain, as the usual negative feedback of allopregnanolone on the hypothalamic–pituitary–adrenal axis may be abrogated by low allopregnanolone, the latter disrupted in a state of chronic stress and depression [43,44,45].

Understanding the mechanism underlying allopregnanolone and the regulation of PMS/PMDD will allow for identification of a key perturbed metabolic enzyme that can serve as a target for drug development for the alleviation of premenstrual disorders.

### Limitations

The primary limitations surrounding this study are the small sample size used and the focus given to the contributions of altered adrenal/pituitary (T/PRL) hormones on premenstrual disorders due to their predominant occurrence during the late luteal phase of the menstrual cycle. As valid as the design of this study may be, it did not take into consideration that neurological factors could influence the abovementioned disorder. To address these limitations and enhance our understanding of the factors contributing to the premenstrual syndrome, a further prospective study involving larger sample sizes should be designed in which the influence of neurotransmitters and neuroactive steroids on premenstrual disorders are vetted.

## 4. Materials and Methods

### 4.1. Subject Recruitment

Participant and Clinical Interaction Resource (PCIR) at the Meharry Clinical Research Center aided in the recruitment of subjects for this study by putting up flyers, posting advertisements in publications approved by Meharry, and through recommendations from existing subjects in similar studies. Inclusion criteria were as follows: (1) women between 18 and 42 years of age at the luteal phase in their menstrual cycle; (2) not pregnant; (3) absence of major findings from a physical examination; (4) no treatment with antidepressants for at least 10 weeks. Exclusion criteria were as follows: (1) evidence of clinically significant physical disorders; (2) clinically significant diagnosable major psychiatric disorder; (3) alcohol or drug dependency within the previous 12 months.

Women who were interested were given a complete description of the study. Written informed consent was obtained from each participant for a 24 mL blood donation. Study participants were evaluated by Neuropsychiatric Interview for Axis I Diagnostic and Statistical Manual of Mental Disorders, 4th edition, criteria (DSMIV-TR), and the severity of depression was determined by the 17-item Hamilton Rating Scale for Depression (HAM-D). Subjects with scores greater than 19 were included in the depression group, while those with scores lower than 19 were included in the non-depression group. The participants completed the HAM-D questionnaire before donating blood. The HAM-D scores and blood samples were collected when the women were 21–25 days into their menstrual cycle (perimenstrual stage). A total of 25 participants (22 black and 3 white, average age of 29.4 years and average weight of 199.8 lbs.) with symptoms of PMS/PMDD were recruited for this study. This study was approved by the Institutional Review Board (IRB) at Meharry Medical College.

### 4.2. HAM-D Scores

The HAM-D score of each woman was calculated based on her answers to the HAM-D questionnaire as previously described [46] (where 0 = absent). Briefly, depressed mood (0–4), difficulty encountered in work activities (0–4), agitation (0–2), insomnia early (difficulty in falling asleep; 0–2), insomnia middle (complains of being restless and disturbed during the night; 0–2), insomnia late (waking up in the early hours of the morning, unable to fall asleep again; 0–2), psychological anxiety (0–4), somatic anxiety (gastrointestinal, indigestion, cardiovascular, palpitation, headaches, respiratory, and genitourinary, etc.; 0–4), and somatic symptoms (loss of appetite, heavy feeling in the abdomen, and constipation; 0–2) [47].

### 4.3. Blood collection and Isolation of Mononuclear Leukocytes

Vacutainer cell preparation tubes with sodium citrate (CPT; BD Biosciences, San Jose, CA, USA) were used to collect 24 mL of blood from each participant using the venipuncture technique. Isolation of mononuclear leukocytes from individual blood samples was performed according to the manufacturer’s instructions, as detailed in Smith et al. [27]. Individual post-centrifugation plasma samples were recovered and stored at −80 °C until used for radioimmunoassay (RIA). Subsequently, mononuclear leukocytes from individual blood samples were harvested and subjected to centrifugal wash as described in Smith et al. [27]; the resulting pellets were stored at −80 °C until used later.

### 4.4. Enzyme-Linked Immunosorbent Assay (ELISA)

Proteins were extracted from each PBMC sample using 200 μL of Pierce RIPA Buffer (Pierce BioTechnology; Rockville, IL, USA) containing a Halt protease inhibitor cocktail. After high-speed centrifugation for 15 min, the supernatant was collected and stored at −80 °C for subsequent analysis by ELISA. The ELISA protocol from the manufacturer (Human Arrestin β1: ARR β1 ELISA kit; MyBioSource, San Diego, CA, USA), which is detailed in Smith et al. [27], was used to determine the levels of β-Arrestin1 protein. Briefly, 10 µL of balance solution and 50 µL of the conjugate were added to each well, except for the blank control well. The plate was mixed using a plate shaker for 1 h at 37 °C. The plate was then washed five times with 1× wash solution. Then, 50 µL of each substrate A and B were added to each well. The plate was covered and incubated for 15 min at 37 °C. Finally, 50 µL of stop solution was added to each well and the plate was read at 450 nm. The β-Arrestin1 protein level in each sample was determined using a standard curve constructed for each assay. The protein concentrations of each sample were determined using the BCA assay (Pierce BioTechnology; Rockville, IL, USA, product #23225). The β-Arrestin1 protein level in each sample was then calculated based on β-Arrestin1 level as ng per mg of protein. The standard curve and the % CV for the duplicate samples were all within the acceptable range for each assay.

### 4.5. Radioimmunoassay (RIA)

Plasma concentrations of T and PRL were measured by RIA methods in the Meharry Medical College Endocrine Core laboratory (MMCECL). The assay sensitivity for T in MMCECL is 0.06 ng/tube and the intra-assay coefficient of variation (CV) is 7.8%. The assay sensitivity for prolactin (PRL) in MMCECL is 0.1 ng/tube and the intra-assay CV is 3.0%. Because all samples were run in one individual assay, the inter-assay CVs were not warranted.

### 4.6. Statistical Analysis

All calculations and statistical analyses were carried out using Excel Statistical Software (Version 2309) (Microsoft Corporation, USA) and GraphPad Prizm software (Version 10.1.0) (Graph-Pad Software, Inc., San Diego, CA, USA). All group data were tested for significance with unpaired student’s *t*-tests and the relationship between two variables were predicted using linear regression. Because of the clustering of PRL in the upper and lower segments of the scatterplot, the relationship between PRL and HAM-D scores in the upper and lower segments of the scatterplot were each predicted using Pearson and Spearman regression. Results were considered significant if they carried a *p* < 0.05 probability value. Results are expressed as mean ± standard error.

## Figures and Tables

**Figure 1 ijms-24-15449-f001:**
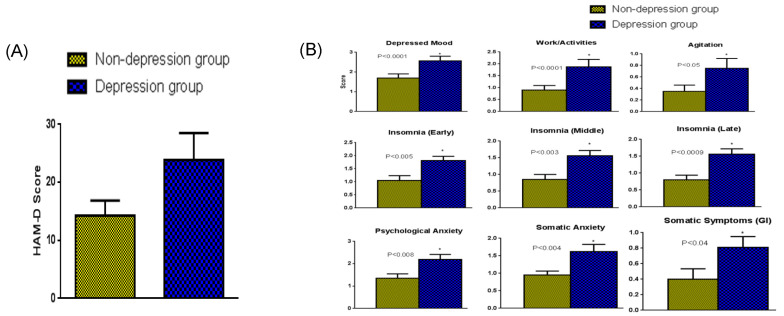
(**A**) The HAM-D scores (mean + SEM) among women in the depression (HAM-D > 19) and non-depression (HAM-D < 19) group. Women in the depression group had a higher HAM-D score (*p* < 0.001) than women in the non-depression group. (**B**) Analysis of the different parameters on the HAM-D scale. Women in the depression group had significantly higher HAM-D scores in all categories tested than women in the non-depression group. * Significantly different.

**Figure 2 ijms-24-15449-f002:**
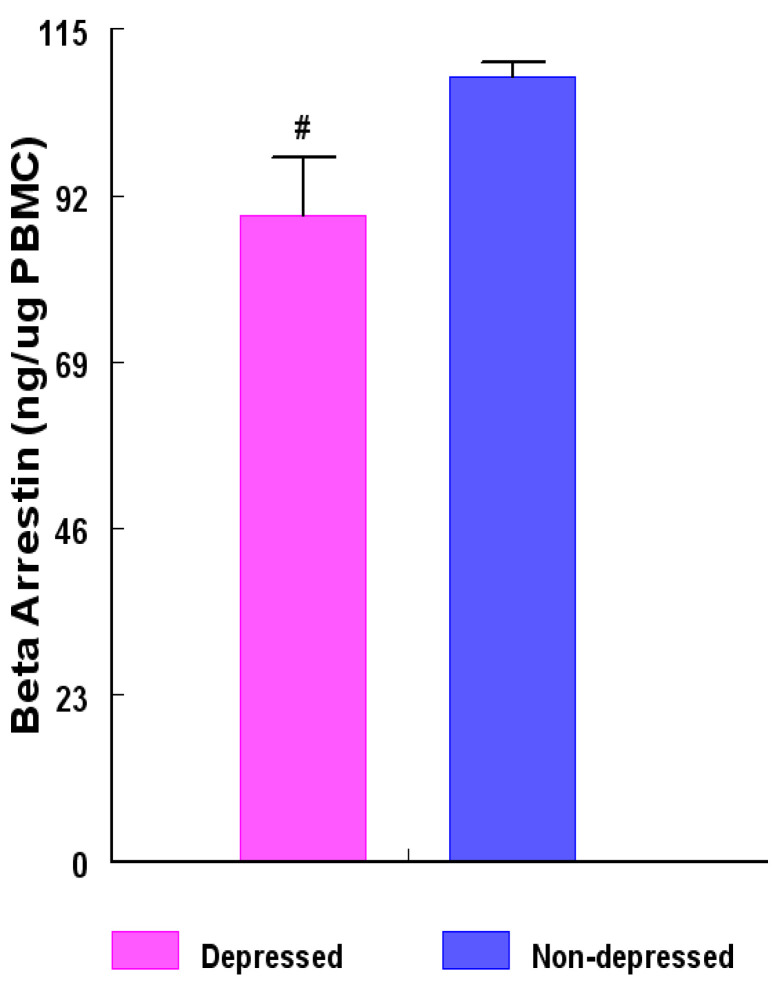
Concentrations (ng β-Arrestin1 protein/µg PBMC) among women in the depressed versus non-depressed group. β-Arrestin1 protein concentrations were determined by enzyme-linked immunosorbent assay (ELISA) and the results between the two groups compared with unpaired ‘*t*’ test. β-Arrestin1 protein concentrations were lower (**#** = *p* < 0.004) in the depression group versus non-depression group. Data are presented as mean ± SEM.

**Figure 3 ijms-24-15449-f003:**
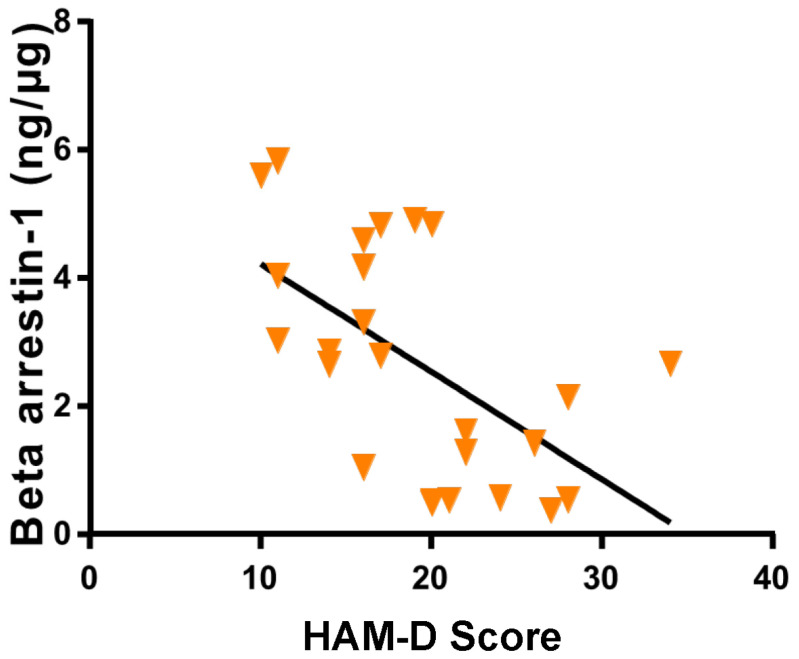
Variation in β-Arrestin-1 concentrations in research subjects related to HAM-D score. A negative correlation was observed (R = −0.63) between β-Arrestin-1 levels and HAM-D score (*p* < 0.0009; 95% confidence interval).

**Figure 4 ijms-24-15449-f004:**
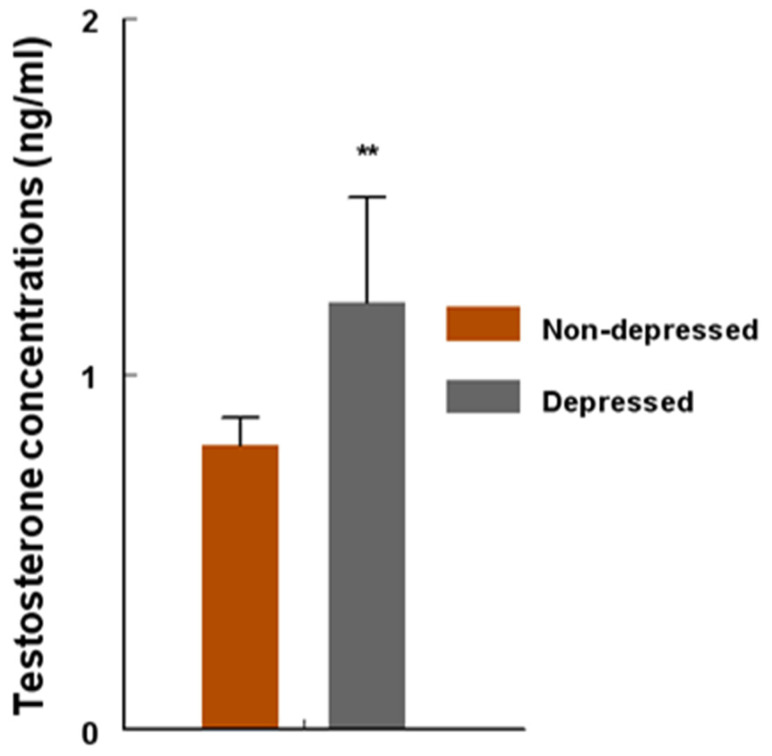
Concentrations (ng/mL) of circulating testosterone (T) among women in the non-depression versus depression group. T concentrations were determined by radioimmunoassay (RIA) and compared with unpaired ‘*t*’ test. T concentrations were higher (** = *p* < 0.001) in the depression group versus non-depression group. Data are presented as mean ± SEM.

**Figure 5 ijms-24-15449-f005:**
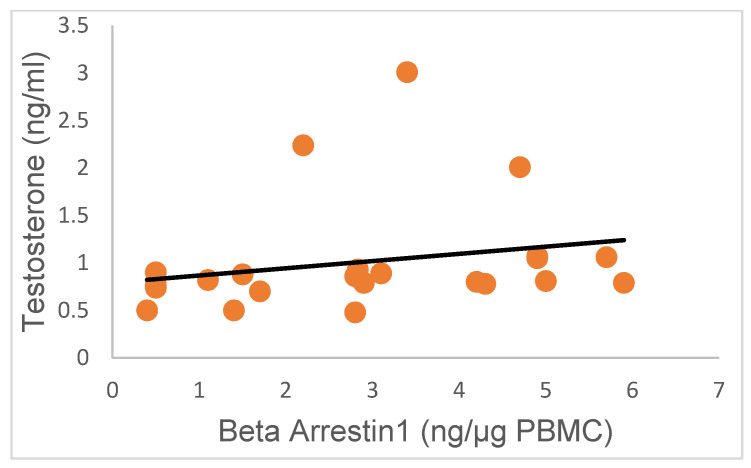
Variation in T concentrations in research subjects relative to β-Arrestin-1 protein concentrations. The concentrations of T were not correlated with the levels of β-Arrestin-1 protein secreted by PBMC (R = 0.22; *p* > 0.05; 95% confidence interval).

**Figure 6 ijms-24-15449-f006:**
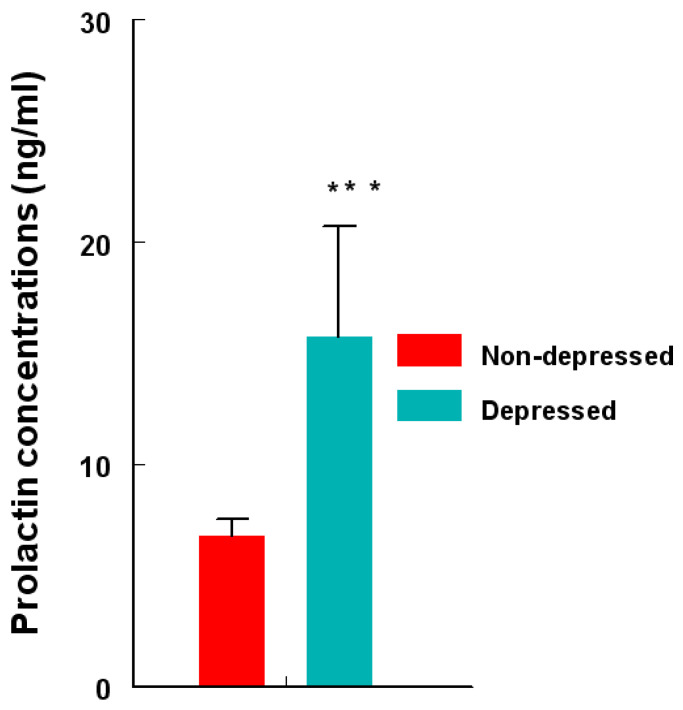
Concentrations (ng/mL) of circulating PRL in depression versus non-depression groups. PRL concentrations were determined by RIA and compared using an unpaired ‘*t*’ test. PRL levels were higher (*** = *p* < 0.0001) in depression versus non-depression women. Data are presented as mean ± SEM.

**Figure 7 ijms-24-15449-f007:**
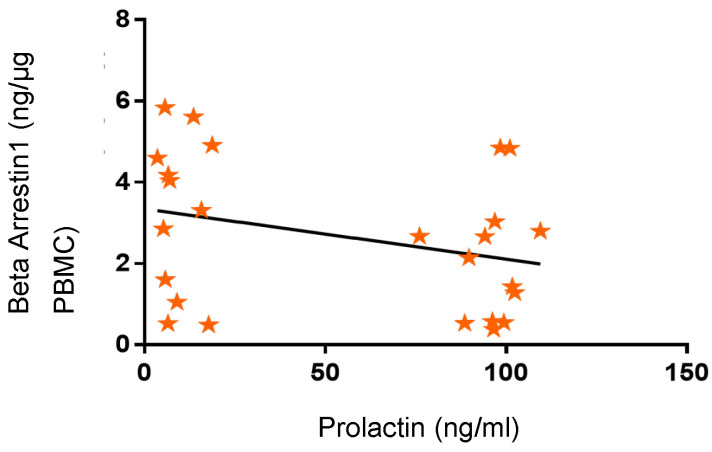
Variation in β-Arrestin-1 concentrations in research subjects related to prolactin (PRL) concentrations. β-Arrestin-1 concentrations were not correlated with PRL concentrations (R = 0.19; *p* > 0.05; 95% confidence interval).

**Figure 8 ijms-24-15449-f008:**
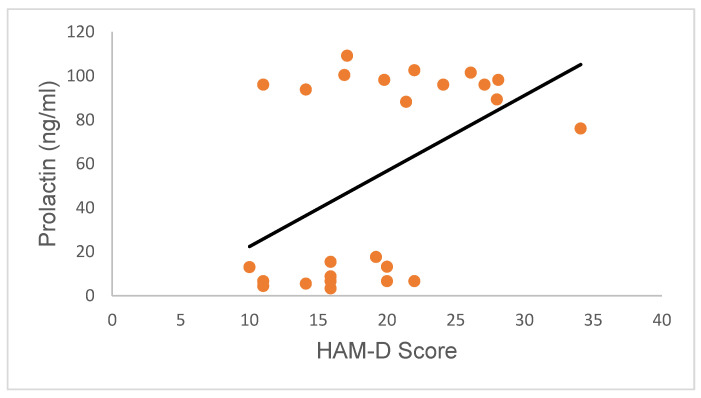
Variation in PRL concentrations in research subjects relative to HRSD scores. PRL concentrations were not correlated with HAM-D scores (upper segment: Pearson R value = −0.5; *p* = 0.08; Spearman R value = −0.26; *p* = 0.39; lower segment: Pearson R value = 0.19; *p* = 0.55; Spearman R value = 0.27; *p* = 0.42; confidence interval for all regression analyses was 95%).

## Data Availability

The data that support the findings of this study are available from the corresponding author upon reasonable request.

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
