# Peer review of "Testosterone and Prolactin Perturbations Possibly Associated with Reduced Levels of β-Arrestin1 in Mononuclear Leukocytes of Women with Premenstrual Dysphoric Disorder"

_ijms, 2023, doi:10.3390/ijms242015449_

Round 1

Reviewer 1 Report

The authors describe associations between depression scores and the levels of B-Arrestin, testosterone, and prolactin among 25 women with PMS in the luteal phase of their menstrual cycle. While interesting, the data are not novel and further description of the cohort and their hormone levels are needed to raise confidence that this could become a significant contribution to the literature as a step towards improved PMDD management.

1.    The introductions is generally well written, but is lacks justification for the assessment of the two hormones, testosterone and prolactin, that are used to test the hypothesis that reduced B-Arrestin is associated with “changes in regulators for premenstrual P4 secretion”.

2.    The contemporaneous levels of progesterone, estrogen and serotonin should be provided, if available. If not available, that should be described as a weakness.

3.    Provide relevant demographic information for the participants, including age and BMI.

4.    Note that Figures 1 and 2 were previously published by MDPI (PMID: 26703643).

5.    Note that all women were diagnosed with PMS, as described in prior publication.

6.    What does the sentence on lines 135-136 mean? It appears T levels were about 50% higher in those with depression, not 98% higher.

7.    Provide the results of linear correlation between B-Arrestin and Testosterone.

8.    Figure 6 shows significant clustering of prolactin levels and simple linear regression is not a valid analytic method.

Sufficient for review

Reviewer 2 Report

Here the measurement of arrestin was performed in the blood of women with Premenstrual Dysphoric Disorder however the study is not demanding. The aim is interesting.

1 describe RIA assay in detail

2. provide study limitations

3. how these results can be important in the understanding of mecchanisms of  PCOS

Reviewer 3 Report

Review Report

Title: 

Reduced Levels of β-Arrestin1 in Mononuclear Leukocytes of Women with Premenstrual Dysphoric Disorder are Associated with Increased Levels of Plasma Testosterone and Prolactin

Authors: 

Sanket Nayyar, Anthony Archibong, and Tultul Nayyar

Comments and Suggestions for Authors:

1. Title Clarity:

   - The title is descriptive and provides a clear indication of the study's focus. However, consider shortening it for brevity while retaining the essence.

2. Abstract:

   - The abstract provides a concise overview of the study's objectives, methods, and findings. It might be beneficial to include a brief statement about the study's implications or significance in the field.

3. Introduction:

   - The introduction effectively sets the context by discussing the prevalence of menstruation-related disorders. However, it would be helpful to provide more background on β-Arrestin1's role in other physiological or pathological processes.

4. Methodology:

   - The participant selection criteria are clear. However, consider providing more details on the methods used to measure β-Arrestin1 protein levels and hormone concentrations.

5. Results:

   - The presentation of results is clear. Consider using more visual aids, such as charts or graphs, to illustrate the differences between the depression and non-depression groups more vividly.

   - It might be beneficial to provide more statistical details, such as effect sizes or confidence intervals, to give readers a better understanding of the study's robustness.

6. Discussion:

   - The discussion provides a comprehensive interpretation of the results. However, it would be beneficial to delve deeper into the potential mechanisms underlying the observed associations.

   - Consider discussing the study's limitations more explicitly, such as potential confounding factors or biases.

7. References:

   - Ensure that all references are up-to-date and relevant to the study. It might be beneficial to include more recent studies on the topic, if available.

8. General Suggestions:

   - Consider conducting further analyses to determine if there are other potential mediators or moderators in the relationship between β-Arrestin1 protein levels and PMDD.

   - It would be beneficial to discuss the potential clinical implications of the findings, especially concerning therapeutic interventions for PMDD.

   - Given the study's findings, consider exploring the potential role of other proteins or hormones in PMDD in future research.

In conclusion, the study provides valuable insights into the relationship between β-Arrestin1 protein levels and PMDD. With some refinements and further elaboration in certain sections, this research can make a significant contribution to the field.

Review Report

Title: 

Reduced Levels of β-Arrestin1 in Mononuclear Leukocytes of Women with Premenstrual Dysphoric Disorder are Associated with Increased Levels of Plasma Testosterone and Prolactin

Authors: 

Sanket Nayyar, Anthony Archibong, and Tultul Nayyar

Comments on the Quality of English Language:

1. General Observation:

   - The manuscript is generally well-written with a clear structure. The language used is appropriate for a scientific audience.

2. Title:

   - The title is clear and grammatically correct. However, consider revising for brevity without losing the main focus.

3. Abstract:

   - The abstract is concise and coherent. Ensure that all sentences are complete and that there are no fragmented statements.

4. Introduction:

   - The introduction flows well. However, some sentences might benefit from slight restructuring for clarity and conciseness.

5. Methodology:

   - The methodology section is clear in its description. Ensure that all technical terms are defined upon first use.

6. Results:

   - The results section is mostly clear. However, consider revising any complex sentences to ensure they are easily understood by a broad scientific audience.

7. Discussion:

   - The discussion is comprehensive. Ensure that all statements are clear and avoid using passive voice excessively.

8. General Language Suggestions:

   - Ensure consistent use of tenses throughout the manuscript. For instance, the methods section should be in the past tense, while general statements should be in the present tense.

   - Avoid using jargon or overly technical terms without explanation.

   - Ensure that all abbreviations are defined upon first use.

   - Check for any typographical or spelling errors that might have been overlooked.

   - Consider having a native English speaker or professional editing service review the manuscript for any subtle language nuances or idiomatic expressions.

Overall, the quality of the English language in the manuscript is commendable. With minor revisions and attention to detail, the manuscript can achieve a high standard of linguistic clarity and coherence.

Round 2

Reviewer 1 Report

The manuscript continues to suffer from lack of citation of the prior studies and lack of important new information.

Adequate for review
